# Occurrence and Population Density of the Endemic Species *Cordulegaster buchholzi* (Anisoptera: Cordulegastridae) on the Cyclades Islands in Greece

**DOI:** 10.3390/insects14110896

**Published:** 2023-11-20

**Authors:** Otakar Holuša, Kateřina Holušová

**Affiliations:** 1Department of Environmental Science and Natural Resources, Faculty of Regional Development and International Studies, Mendel University in Brno, Tř. Gen Píky 7, CZ-613 00 Brno, Czech Republic; 2Department of Forest and Wood Product Economics and Policy, Faculty of Forestry and Wood Technology, Mendel University in Brno, Zemědělská 3, CZ-613 00 Brno, Czech Republic; holusova.katerina@seznam.cz

**Keywords:** *Cordulegaster buchholzi*, Cordulegastridae, Odonata, distribution, population abundance, biogeography, Cyclades Islands, Greece

## Abstract

**Simple Summary:**

*Cordulegaster buchholzi* is an endemic species of southern Greece with a centre of occurrence on the Cyclades Islands. Detailed research on the population of the species was carried out on the islands of Andros, Tínos and Náxos between the years 2010–2023. Based on larval findings, occurrence was found in only 19 streams on these islands. The species inhabits sloping, incised streams in narrow rocky valleys, the beds of which are stony, often even bouldery or with rocky banks. Streams are surrounded by forest belts or a segment of forests dominated by *Platanus orientalis* and *Alnus glutinosa*. The biotopes are strongly influenced by human activity; there is a significant impact of water use for irrigation, pollution from grazing cattle excrement, potential deforestation and forest fires. The range of the species (i.e., extend of occurrence) is only 100 km^2^; the species is therefore among the rarest species of European dragonflies.

**Abstract:**

Our research was focused on determining the geomorphological characteristics of streams, characteristics of sediment in streams, habitat, emergence sites and flight period. Larvae were recorded in 19 streams (altitude of 35–680 m a.s.l.), with an average minimum width of 44.2 cm, an average maximum width of 352.9 cm, an average minimum depth of 9 cm and an average maximum depth (in pools) of 55 cm, with an average stream gradient of 12 grades (range 0.6–45 grades). In terms of grain size, the sediment in these biotopes can be characterized as sandy gravel, medium-grained gravel with an admixture of fine sand and an admixture of coarse-grained gravel prevails (with dominancy of fraction 2–5 mm with a representation of 47%). The larval density reached 0.1–62.2 larvae per 1 m^2^ of suitable sediment. Exuviae (100 exuviae found in total) occurred at an average of 66 cm horizontal distance from the shore and an average vertical height of 124 cm above the ground. The average total distance of larval movement was 190 cm. The emergence site was categorized as larvae-dominated tree trunks (57% of cases), rocks (51%) and overhanging rocks (11%). The flight period was recorded from 17th May to 15th July (literary record—to 15th August) with peak flight activity noted in the third quarter of June. Considering the size of the area—extent of occurrence, the population of *C. buchholzi* is strongly threatened; according to the IUCN categories it should be classified as endangered (EN).

## 1. Introduction

*Cordulegaster buchholzi* was described by Lohmann [1] within the description of the species *Cordulegaster helladica* as subspecies *Cordulegaster helladica buchholzi* with material from the Cyclades Islands (locality Naxos, Andros) in Greece. The species *Cordulegaster helladica* occurs at the southernmost part of the Balkan Peninsula—the Peloponnés peninsula, in the area from Mt. Parnassos after Attica, the island of Euboia and the Cyclades Islands [2,3]. The subspecies *C. helladica buchholzi* was reported only from the islands of Andros, Naxos and Tínos [2,4,5,6,7,8,9,10]. Most recently, however, Schneider et al. [11] revised the taxonomic position of this subspecies on the basis of genetic analysis, where they established *Cordulegaster buchholzi* (Figure 1) as a species and moved the limits of occurrence towards central Greece—the Euboia peninsula and the Attica region. As the boundary of the distribution remains unclear, the range of the species is currently unclear, whether it is limited only to the Cyclades Islands or includes part of mainland Greece.

There is very little knowledge about the ecological risks of the species, its exact distribution and threats. Since its description in 1993 [1], data on its occurrence have been increasing, a fact which was already taken into account in atlases of the distribution of dragonflies within Greece [6], the Mediterranean [12] or Europe [2]. By raising it to the status of a species [11], questions arise regarding not only about the limit of distribution, but also about the ecological requirements of the species and its potential threat, since it is a dragonfly species with a very small area (extent of occurrence) of distribution limited to an area of several hundred km^2^.

The aims of this paper are to make a complete overview of all findings of *Cordulegaster buchholzi* on the Cyclades Islands in Greece that were found during intensive research in the period of 2010–2023, to make a first evaluation of population abundance and to evaluate stability. The knowledge on occurrence and ecology of this species is very important also from the perspective of listing the subspecies of *Cordulegaster helladica* as “EN” in the European Red list of Dragonflies [13]; therefore, evaluation of *C. buchholzi* as a separate species based on current data on distribution, population trend and its threat, is very real.

## 2. Materials and Methods

Detailed surveys in the area of the Cyclades Islands were carried out on Andros island—in the periods 10–15 June 2010; 12–20 October 2013; 22 June–3 July 2014; 9–15 October 2016; 11–18 June 2018; 4–16 July 2021; 26 July–2 August 2022 and 28 May–12 June 2023—at Tínos island—in the time period 11–18 October 2014—and at Naxos island—in the periods 4–13 October 2017 and 10–13 July 2021.

All available watercourses on the three islands were explored—Andros, Tínos and Naxos. Watercourses on Tínos and Náxos islands were surveyed only once. Watercourses on Andros Island were repeatedly surveyed. By finding larvae in 2010 in the northern area of Andros, streams in the watershed of Aspropotamos stream were selected for intensive surveys. Several sites were then selected on each stream to survey for larvae in detail, with individual sites located approximately 500–700 m from each other depending on the nature of the stream. A number of sites were then selected on streams (Table 1). Only sites where *Cordulegaster buchholzi* was detected are listed in the site descriptions.

In all watersheds, individual sites (locations) were selected on the main stream and tributaries where larvae were surveyed over a 50 m transect. In this reach, all suitable sediment deposits where larvae were suspected to occur were examined in detail. Where larvae were found, basic stream characteristics (stream width—minimum/maximum), stream depth (minimum/maximum in pools) and stream gradient were recorded. Phytocenological notes were made on vegetation and tree cover composition, and other dragonfly species were monitored. All larvae found were after registration again released into the stream. At the site where larvae were detected, a sediment sample weighing approximately 700–1000 g was collected to determine the sediment grain fraction representation. Grain size was determined on the dried sample, which was divided into fines and skeleton fractions. The skeleton was separated into grain size fractions (2–5 mm, 5–20 mm and 20 and more mm) using sieves and the fine earth into grain size fractions (less than 0.05 mm, 0.05–0.1 mm, 0.1–2 mm) using the floating method [14]. For the 50 m transect, the suitable sediment area was estimated given by the average length and average width.

Emergence monitoring was conducted on all streams. 100 m transect was selected where larvae were detected. Exuviae were collected in the riparian parts of the sites studied. They were sought up to about 10 m from the shoreline and up to 5 m on tree trunks. The distance from the shoreline (vertical projection of the emergence place) and the height above the ground (vertical distance from the place where the shoreline ends) were measured. The total distance to the emergence place is the sum of the height and the distance from the shore which is the total distance travelled by the larvae from the shoreline to the place where the imago hatched. As position of the exuvia, the position of its thorax was used. Distances were measured from this point. The type of emergence place was evaluated according to place, where exuvia were attached—tree trunks, rocks, overhanging rocks, stone walls and bridges.

In 2018, 2021, 2022 and 2023 the flight period of the imagines at the Aspropotamos stream in Remata place (sometimes called Revmata or Remmata, Arni village, locality VIIe) was studied in detail. The flight activity of imagines was monitored in suitable weather (partly cloudy to clear sky, midday temperature around 18–20 °C or more) from 6:30 a.m. to 7 p.m. EET in the periods from 11st to 18 June 2018, from 4 to 16 July 2021, from 26 July to 2 August 2022 and from 28 May to 12 June 2023. Sunrise on days at this latitude in mid-June is at 5:50 a.m. EET, and at second half of July from 6:15 a.m. EET. To evaluate the flight activity, the imago was marked, but for the overall evaluation, all flights were summarized, whether it was a repetitive flight was not taken into account.

The distribution of the species on the Cyclades islands in Europe was processed on the basis of the occurrence of larvae, i.e., our data were used (Figure 2). Literary data in the maps were not taken into account due to difficult localization or accidental finding of imagines. Our own data are compared with literary knowledge from the literature in the discussion.

Data and statistical evaluation were processed in software MS Excel (Microsoft Professionals 2016, Redmond, Washington, DC, USA), the maps were processed in ESRI 2020 ArcGIS ArcMap 10.8 software (Microsoft, Redmond, Washington, DC, USA).

## 3. Results

### 3.1. Area of Species at Cyclades Islands in Greece

*Cordulegaster buchholzi* was found at 32 localities (Table 1) in 19 streams, i.e., 18 stream catchments—at 22 localities on Andros, at 3 localities on Tínos and 7 localities on Náxos island. Localities at Andros are concentrated in the central part of the island and centred around the highest massif of Petalo Mt. On the island of Tínos, these localities are in the central part of the island. On the island of Náxos, localities are concentrated in the northern quarter of the island (Figure 2). The localities lie at altitudes from 35 to 680 m a.s.l., the average altitude is 272 m a.s.l. (Figure 3).

Due to the current nature of all the islands of the Cyclades, which have been deforested in history and the majority of the landscape is currently used for shepherding, the localities are concentrated in forested areas, for example the valley of the Aspropotamos stream in the vicinity of Remata on Andros island, which is one of the most forested areas on the Cyclades (Figure 4a), or on more deforested islands, i.e., Tínos (Figure 4b) and Náxos, the localities are located in incised valleys where belts of forest vegetation or belts of trees along watercourses are preserved.

The area of occurrence of *Cordulegaster buchholzi* at Cyclades Islands in Greece, i.e., extent of occurrence, is approximately 100 km^2^ (the sum of the area of the islands), while the localities themselves are found only in the central or northern parts of the islands with the occurrence of trees or forest belts, i.e., area of occupancy reaches only 0.2 km^2^. The area constitutes three areas within a disjunctive area, with distance of 20 km (distance from the locality in the south of Andros—locality in the northwest of Tínos) and of 50 km between individual parts (distance between the locality in the east of Tínos—the westernmost locality on the island of Naxos). In these areas there is either the sea or there are deforested forestless parts of land without any suitable habitat.

### 3.2. Habitat Characteristics

The habitat of the species are streams in forests or with belt of forest vegetation with an average minimum width of 44.2 cm, an average maximum width of 352.9 cm, an average minimum depth of 9 cm and an average maximum depth (in pools) of 55 cm (Figure 5a), with an average stream gradient of 12 grades (range 0.6–45 grades). The presence of suitable sediment is necessary for the larvae to survive, as is common with all *Cordulegaster* species. In biotopes, the sediment is formed by the weathering of clasts and phyllites, or gneisses, which are the subsoil in the area of the species’ biotope. According to grain size characteristics (Figure 5b) with an average fraction less than 0.05 mm 0.4%, fraction 0.05–0.1 mm with 1.3%, fraction 0.1–2 mm with a representation of 28.3%, fraction 2–5 mm with a representation of 47%, fraction 5–20 mm with a representation of 14.5% and fraction 20 mm+ with a representation of 8.5%. Thus, the ratio of fines (fraction less than 2 mm grain size) to skeleton (fraction greater than 2 mm) averages 25.5, 74.5% in the sediment. It is evident from this that the sediment in these biotopes is sandy gravel, dominated by medium medium-grained gravel with an admixture of fine sand and an admixture of coarse-grained gravel. Stones and boulders (from 20 mm) are very common in riverbeds.

The banks are steep, sometimes even vertical, often rocky. All streams are semi-natural (Figure 6a,b and Figure 7a,b), there is significant human influence on all streams. All streams are more or less straight, stony, even bouldery, sections with rocks and deep pools in the upper reaches of watercourses are frequent (Figure 6b). 

Suitable sediment for larvae can be found in coves, behind stones, or in small lateral pools (Figure 8a). Due to the dynamics of water flows during the year, high water levels are evident in the riverbeds (the level in the spring months after intense rains reaches up to 200 cm higher). In some parts, the banks of the streams are reinforced with stone walls, or water pipes are excavated in the rocks. Currently, hoses or pipes are installed here to drain water to agricultural land in the vicinity (Figure 8b). Stream alluvium is covered by a vegetation cover with average cover of 30% (dependent on tree cover, cover varying from 5 to 90%).

The bank vegetation is most frequently composed of (in order of frequency of occurrence)—*Rubus ulmifolius*, *Equisetum telmateia*, *Carex pendula*, *Mentha longifolia*, *Urtica dioica*, *Hedera helix*, *Cyclamen purpurascens*, *Athyrium filix femina*, *Lysimachia* sp., *Asplenium* sp. and *Polypodium vulgare* are common on stones or rocks in shaded places.

The alluvia are covered by forest stands or belts of trees with composition—*Platanus orientalis* and *Alnus glutinosa* (90% of forest belts and forests) (Figure 6 and Figure 7). The following tree species occur individually: *Olea europea*, *Nerium oleander, Ficus carica*, *Morus alba*, *Acer monspessulanum*, *Cupressus sempervirens*, *Populus nigra*, *Quercus ilex* or *Q. coccifera*.

In habitats with *Cordulegaster buchholzi*, six other dragonfly species were found. Three species were abundantly occurring—*Calopteryx virgo festiva*, *Platycnemis pennipes nitidula*, other species *Calieschna microstigma*, *Onychogomphus forcipatus*, *Orthetrum anceps* and *Chalcolestes parvidens* were always detected, but in individual specimens or larvae, up to a maximum of 10 individuals.

### 3.3. Abundance of Population

Larvae occurred in areas with suitable sediment (Figure 8a), which occurs in shallow parts of the bed (e.g., locality VIIe) or also occurs in deep pools (e.g., localities Ia, Ib). On average, larval abundance (all instars combined) was 8.3 larvae/50 m section. With respect to sediment area, the density reached 0.1–62.2 larvae per 1 m^2^ of sediment (Table 1). The maximum number of larvae found per 50 m section of watercourse was 70, with a maximum of 25 larvae per site (50 × 35 cm). The high density values of larvae at localities VIIg and XVIIb were caused by the crowding of a large group of larvae into one pool due to the local suitability of the pool, as the other part of the riverbed was without sediment and therefore larvae did not occur there. Larvae were also found in the spring parts of the streams (Figure 8b), where the sediment consisted only of organic material.

Due to the local suitability of the sediments, or the presence of water only in part of the flow, i.e., in springs, there is a clustering of larvae (Figure 1b). In areas where there is a high degree of deforestation, i.e., Tínos island, larvae were also found in streams between agricultural plots with individual trees (Figure 7a), and these biotopes can be classified as remote from nature to completely artificial.

It is clear from the distribution of sites that most larvae were found in the upper parts of the streams that lie in forest vegetation. Larvae found individually in the lower parts of the streams are due to larval flushing during high flow conditions. In these lower parts, the occurrence of imagines (both males and females) was recorded, but no oviposition was detected. Females most often prefer streams with character like those in Figure 6a and Figure 8a.

### 3.4. Time and Place of Emergency

The emergence period was determined to be from the last day of May to 10 July (the latest date of detection of exuviae in 2018). A total of 100 exuviae were found during all research periods (several exuviae were in varying degrees of damage—the abdomen or head were missing); however, fresh exuviae were found only until the middle of June, during the other months of July–October exuviae were found already damaged or with cobwebs or otherwise “stale”. The first individuals, taking into account the detection of adult flight imagines at the end of May, must hatch already in the beginning of May.

The place of emergence was evaluated with respect to their location, i.e., it was the place where the imago hatched: (a) tree trunks, (b) boulders and rocks, (c) overhanging rocks, (d) stone walls or (e) bridges (Figure 9). The “offer” of these emergence places is based on the character of the habitat that the species inhabits, which are watercourses with rocky streambed with boulders bank, with trees at banks. Therefore, tree trunks dominated as a type of emergence place (57%, Figure 9). There are often more exuviae on the trunks, in locality VIIe 25 exuviae were found on one *Platanus orientalis* trunk up to a height of 350 cm above the ground; in some cases there were even three exuviae on top of each other. From other places, larvae use rocks (51%) or overhanging rocks (11%), stone walls (5%) and bridges across streams (3%) (Figure 9). On overhanging rocks, it is obvious that the larva at the end of its “journey” was climbing the ceiling as it were and moving horizontally suspended from the ceiling.

Emergence places were at different distances from the shoreline and at different heights—exuviae occurred at a horizontal distance from shore 0 to 360 cm (ẍ = 66 cm) at a vertical height above the ground from 12 to 420 cm (ẍ = 124 cm), the total distance larval movement was from 12 to 495 cm (ẍ = 190 cm) (Figure 10 and Figure 11). The vast majority of exuviae were found within 200 cm of the shores, the larvae had no tendency to climb to greater distances (Figure 12); on the contrary, 30% of the exuviae were at zero distance from the shores.

### 3.5. Flight Time of Imagines

The flight period of imagines was observed to be from 28 May to 15 July; no flight imagines were detected in the last week of July (Figure 13). On the 28th, a fully mature imago was found, it can be assumed that it had already been “in flight” for a week, i.e., that hatching can be dated to the first half of May. When including the dated literature data [1,4,6,7,9,15] flight activity was detected from 17 May to 15 August and peaks in the second half of June. The end of the flight season probably falls in the second half of July, the individual detected in August can be considered exceptional. However, the frequency of flying imagines does not reach high numbers; in one day in the first half of June, a maximum of 40 overflights of imagines were detected at one location during the entire day of observation.

### 3.6. Threatenig Factors and Conservation Status

The biotope of the species is significantly affected by four negative factors: a. deforestation, b. pollution by excrements of grazing cattle (goats), c. drainage of water from watercourses for irrigation of surrounding land and d. fires. The deforestation factor is a potential one, although the Cyclades Islands have historically experienced extreme deforestation, and forest complexes or forest belts have been preserved only in valleys with watercourses. The faecal pollution factor is very intense and has been recorded in all watercourses, as goats are often hidden there in the midday hours or come there for a source of water to drink. The density of excrement is locally very high. The water drainage factor was recorded in all watercourses, as well as springs (Figure 8b). Water is sometimes collected in concrete tanks. In the case of local draining of time, the water flow can suddenly decrease or even stop (observed at location VIIf, on 12.X.2016). The occurrence of fires, especially in mainland Greece, has been a regular phenomenon in recent years and local fires also occurred on Andros in 2020. Therefore, even forest belts in the valleys can be affected by fires, which could lead to deforestation and significantly affect the abundance of the species.

The area of occurrence of *Cordulegaster buchholzi* in the Cyclades Islands in Greece, i.e., extent of occurrence, is only 100 km^2^, while the area of occupancy reaches, which are the watercourse’s own biotopes, reaches an area of only 0.2 km^2^. Based on these data, the species falls under the IUCN [16]—endangered (EN) category.

## 4. Discussion

Although the species was only described in 1993 [1], its occurrence in the Cyclades, specifically on the island of Naxos, was first mentioned by Karl Buchholz [15], after whom the subspecies was later named. The findings at that time were surprisingly included in the occurrence data of *Cordulegaster insignis* [1,15,17,18]. The occurrence of the *Cordulegaster helladica buchholzi* subspecies is later mentioned in literary records [4,5,7,8,9,10], in the work of Schneider et al. [11] the subspecies was assigned species status.

*Cordulegaster buchholzi* is a European or Greek endemic species with very small area, which includes the Cyclades Islands—Andros, Tínos and Náxos [2,8,19]. If its area were limited to the Cyclades Islands, its area would be only 1000 km^2^. Schneider et al. [11], however, report the occurrence of several individuals in the region of Attica and the southern part of the island of Euboia; the authors’ own findings of *Cordulegaster* sp. specimens in the area of Attica in the wider vicinity of Rafina, however, do not confirm this statement, as the species *Cordulegaster helladica* is widespread here, individuals of which are morphologically different from individuals of *Cordulegaster buchholzi*. In addition, in this area the populations achieve different abundances and different features of flight activity [Holuša unpubl.]. The distribution of the species *C. buchholzi* in the area of Attica and the island of Euboia is still debatable. If it included a part of Attica, even so, the total area would only reach 8000 km^2^ (i.e., extent of occurrence), which would be one of the smallest European areas. With regard to the size of the area and the abundance in the localities, it is among the rarest species of the genus *Cordulegaster* in the western Palearctic.

*Cordulegaster buchholzi* is classified as a Moesic–Thracian faunal element [1], none the less, its area of distribution is probably still limited to the Cyclades Islands, or the Attica region since the glacial period, it would be more appropriate to classify it as Aegean faunal element, i.e., among the species, which in the postglacial period did not significantly expand their range from the Aegean Sea, in the case of *C. buchholzi* remaining limited to a very small area in southern Greece.

On other islands there are currently other species of the genus *Cordulegaster* sp. relatively well known [19,20,21,22,23], it could potentially occur on the nearest islands at the Cyclades archipelago. However, in the Attica region and east of Mt. Parnassos is so far not much data, which is also due to the rarity of aquatic biotopes of flowing waters in this area. Knowledge about the occurrence of *Cordulegaster helladica* is known mainly from the Peloponnés region [24,25,26,27].

*C. buchholzi* inhabits from geomorphological characteristics point of view quite a different type of habitat compared to other species. In contrast to the Central European species, this species inhabits watercourses in the Cyclades, which are formed by boulder to rocky riverbeds, often without undergrowth of vegetation [28,29]. The most similar are the habitats of the species *Cordulegaster picta* in northern Greece [Holuša unpubl.]. They are similar to *Cordulegaster helladica* habitats, although the habitats are more overgrown with vegetation, the bed is not as steep and rocky [Holuša unpubl.]. The tree composition of the habitats of *C. helladica* is identical—the dominance of *Platanus orientalis* and *Alnus glutinosa*, but the habitats are sunnier, as they are not located in such steep valleys [Holuša unpubl.].

From the point of view of the substrate that the larvae seek, there are no data yet for other species occurring in Greece. From the group of *Cordulegaster* species that inhabit a similar habitat, i.e., flowing streams, data are available from Central Europe for the species *Cordulegaster heros* [28,30,31], whose larvae choose sandy-gravel sediment (sand 79.6% and gravel 15.6%) with a small admixture of silt (4.8%) [31], or in Austria [31] the most favourable microbiotope is characterized by fine-, medium- and coarse-sand substrate. From Baden-Württemberg in Germany, in the species *Cordulegaster boltonii*, a sediment composition of 35% sand, 29% clay, clay, 33% gravel and 3% others was found [32].

With regard to larval density, comparable data are again not available for species from Greece. However, the species *C. buchholzi* is characterized by relatively frequent “clustering” of larvae. Similar clustering is described for the species *Cordulegaster heros*, where was recorded the highest local density of 25 larvae per 0.25 m^2^ of sediment (majority of last larval instar—14) in a suitable pool [33], or for the species *Cordulegaster bidentata*, where was recorded the highest local density of 18 larvae per 0.08 m^2^ of sediment (majority of last larval instar—9) in a suitable pool at small forest spring stream [Holuša unpubl.].

The choice of emergence site for this species is of course determined by the microhabitats available on the shores, which then also leads to larvae climbing relatively smooth rocks or climbing overhangs. However, from a comparison of data for the species *Cordulegaster helladica* [Holuša unpubl.], larvae of *C. buchholzi* did not cover such a long distance from the shore with a maximum distance of approx. 500 cm, while larvae of *C. helladica* are able to climb the trunk of *Platanus orientalis* up to a height of 850 cm [Holuša unpubl.].

The species *Cordulegaster buchholzi*, compared to other species of the genus *Cordulegaster* sp. [28,34], is a species with an early start of the flight period and a very short period of flight activity. The first individuals are already known from the middle of May [15], and with a relatively early termination in the last week of July, and findings from the beginning of August are already rare [6]. In contrast to *C. helladica*, which flies even in higher numbers in the first days of August [Holuša unpubl.].

The subspecies of *Cordulegaster helladica helladica* and *Cordulegaster helladica buchholzi* in the European Red list of Dragonflies [13] are included in the “EN” category. The obtained data confirm the inclusion of the *Cordulegaster buchholzi* species in the “EN” category, even if the data on the population trend are not exactly known, as only a few locations were replicated at the locations for the period 2010–2023.

## 5. Conclusions

*Cordulegaster buchholzi* is a Greek endemic species whose range includes the Cyclades Islands—Andros, Tínos and Náxos. Through intensive research in the years 2010–2023 on the Cyclades Islands, the occurrence of the species was found in 19 streams on all three islands. Most sites are concentrated in the forested area in the central part of the island of Andros. The streams, in which the larvae were found, contain sandy-gravel sediment, the stream bed is stony and sometimes steep bouldery to rocky. The flight period of imagines was observed from 28 May to 15 July, when including dated literary data flight activity was detected from 17 May to 15 August and peaks in the second half of June, whereby the species has a slightly shifted flight period to the spring months.

However, it remains unclear whether the species is widespread in Euboia or in Attica in Greece. Even so, it belongs among the rarest species of Europe, or the species of dragonflies of the western Palearctic, with current classification in the Red List of the IUCN category—Endangered (EN).

## Figures and Tables

**Figure 1 insects-14-00896-f001:**
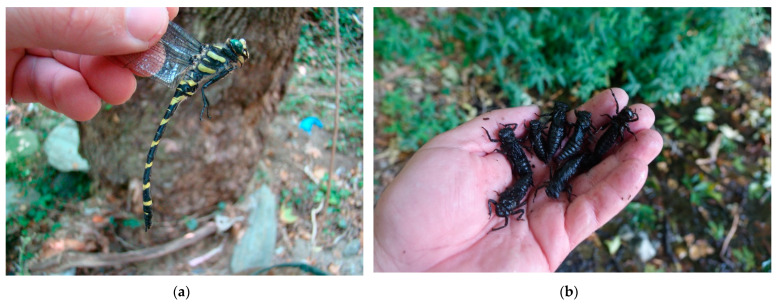
Adult and larvae of *C. buchholzi:* (**a**) male and (**b**) larvae—high density of larvae in the source of the stream.

**Figure 2 insects-14-00896-f002:**
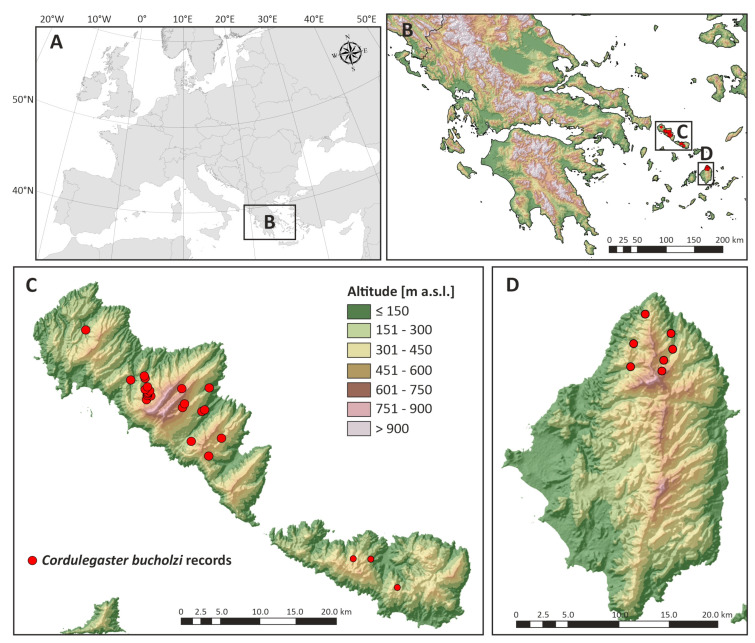
Known records of occurrence of *Cordulegaster buchholzi* at Cyclades Islands in Greece (based on the detected occurrence of larvae, state to 30 August 2023) (**A**) display area within Europe, (**B**) display of occurrence within the territory of Greece, (**C**) display of occurrence within the islands of Andros and Tínos and (**D**) Naxos island.

**Figure 3 insects-14-00896-f003:**
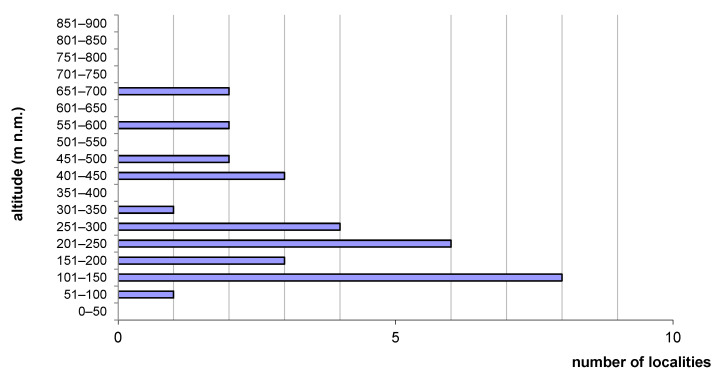
Number of localities depending on altitude with *Cordulegaster buchholzi* occurrence on Cyclades islands in Greece (all authors data (N = 32)).

**Figure 4 insects-14-00896-f004:**
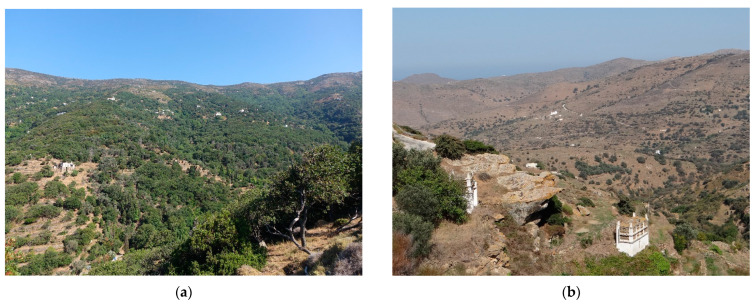
Character of the landscape of islands in the Cyclades Islands with valleys with watercourses: (**a**) Andros Island, valley of Remata and (**b**) Tínos island, valley near Tarampados.

**Figure 5 insects-14-00896-f005:**
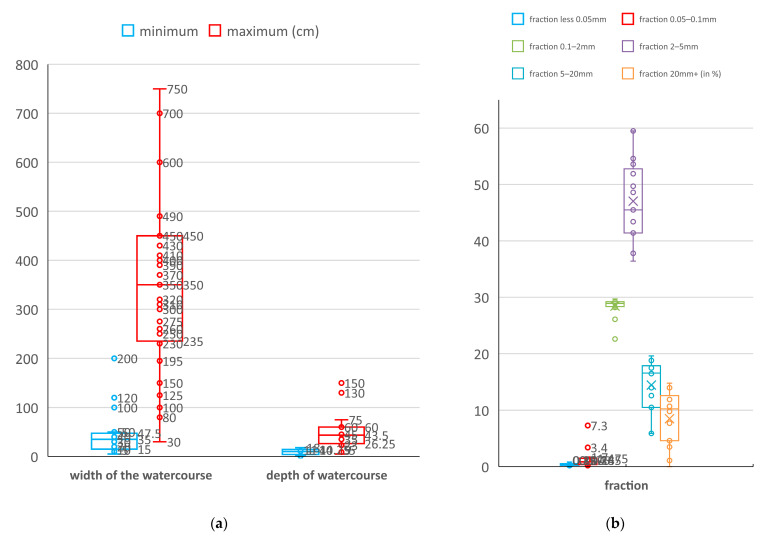
Characteristics of habitat at localities with *Cordulegaster buchholzi* (N = 32): (**a**) width and depth of watercourses and (**b**) grain composition of the sediment with larvae (the box represents the values between the upper and lower quartiles, the whiskers represent the range of values; values beyond the whisker are considered outliers; in the box, the horizontal line is the median, the cross is the mean).

**Figure 6 insects-14-00896-f006:**
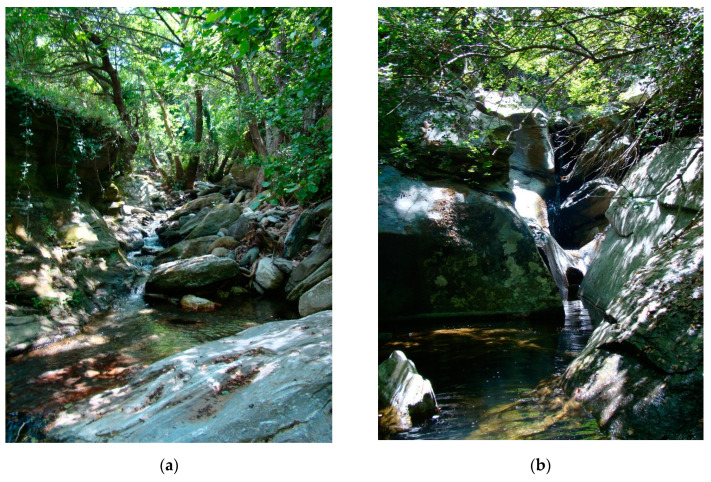
Habitats of *Cordulegaster buchholzi* at Andros Island in the Cyclades Islands in Greece (**a**) locality of Remata—Aspropotamos stream (locality VIIe) (10.VI.2010, photo Otakar Holuša and (**b**) locality of Apikia—Vorimi Spilio (locality Ib) (11.VI.2010, photo Otakar Holuša).

**Figure 7 insects-14-00896-f007:**
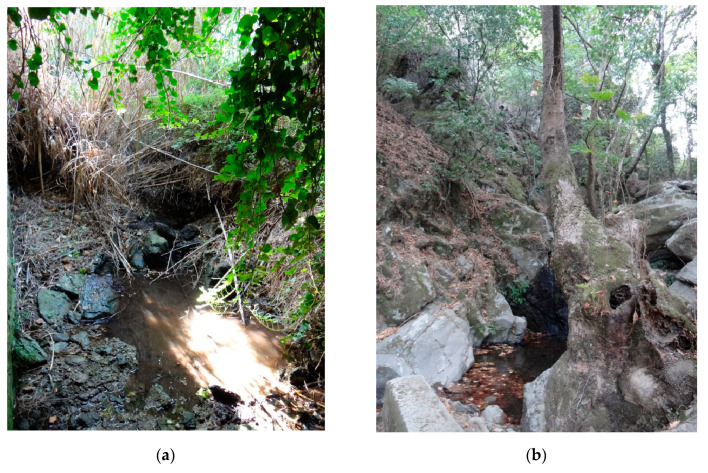
Habitats of *Cordulegaster buchholzi* on Tínos and Naxos in the Cyclades Islands in Greece (**a**) locality of Tarampados (locality XIIIa) (15.X.2014, photo Otakar Holuša) and (**b**) locality of Koronida (locality XVIIa) (5.X.2017, photo Otakar Holuša).

**Figure 8 insects-14-00896-f008:**
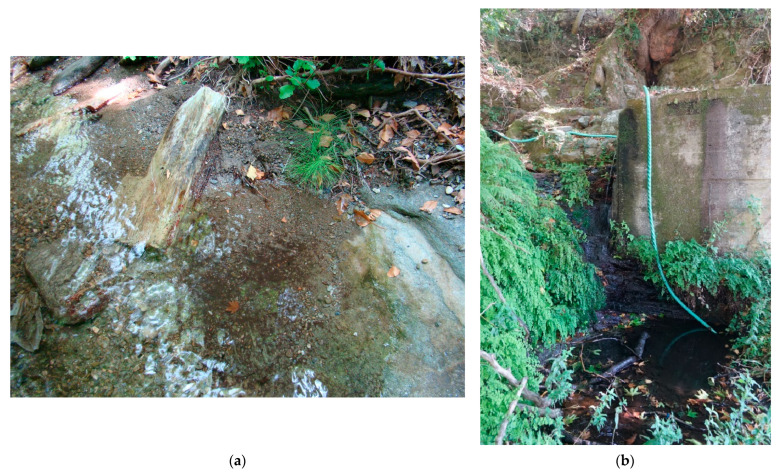
Detail of habitat of *Cordulegaster buchholzi* on Andros Island in the Cyclades islands in Greece (**a**) locality of Remata—Aspropotamos stream (locality VIId) (17.VI.2013, photo Otakar Holuša) and (**b**) locality of Arni (locality VIIa) (31.VII.2022, photo Otakar Holuša).

**Figure 9 insects-14-00896-f009:**
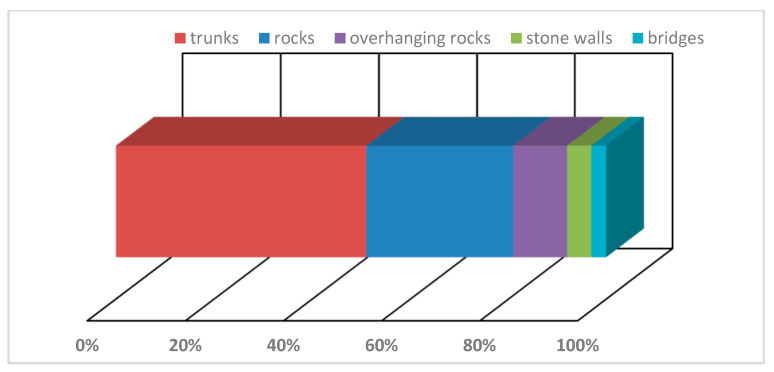
Proportion of different types of emergence place of *Cordulegaster buchholzi*.

**Figure 10 insects-14-00896-f010:**
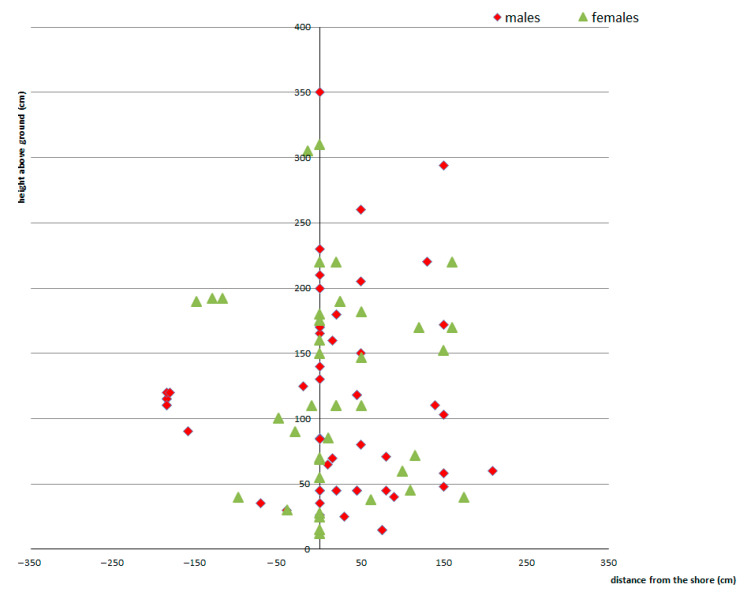
Position of exuviae in riparian parts of the habitat of *Cordulegaster buchholzi* (x-axes: (−) left bank and (+) right bank).

**Figure 11 insects-14-00896-f011:**
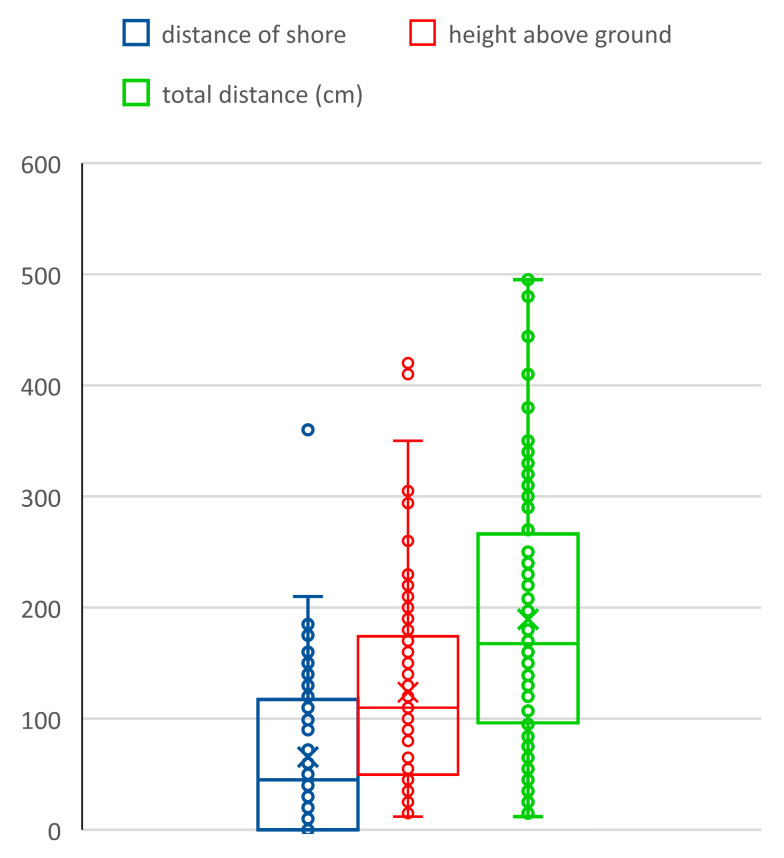
Variability of position of exuviae of *Cordulegaster buchholzi* (distance from shore, height above ground and total distance; all in cm; boxplot description see Figure 5).

**Figure 12 insects-14-00896-f012:**
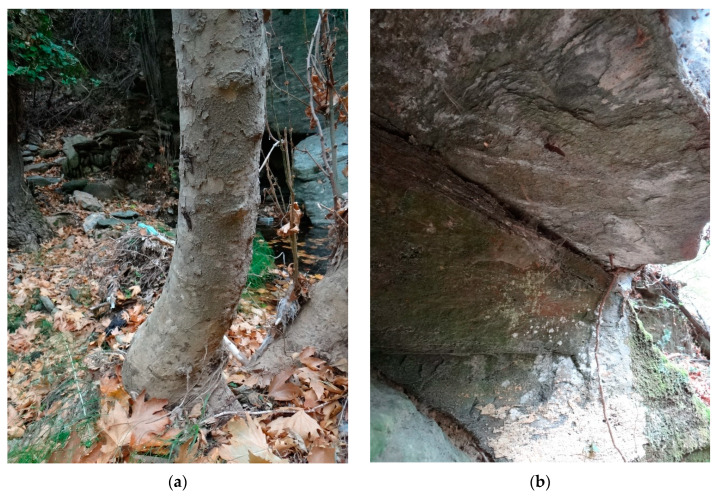
Position of exuviae of *Cordulegaster buchholzi*: (**a**) on trunk of *Platanus orientalis*, locality of Remata—Aspropotamos stream (locality VIIf), (17.X.2013, photo by Otakar Holuša) and (**b**) exuvia hanging suspended on a horizontal slab of rock, Remata—Aspropotamos stream (locality VIIg), (17.X.2013, photo by Otakar Holuša).

**Figure 13 insects-14-00896-f013:**
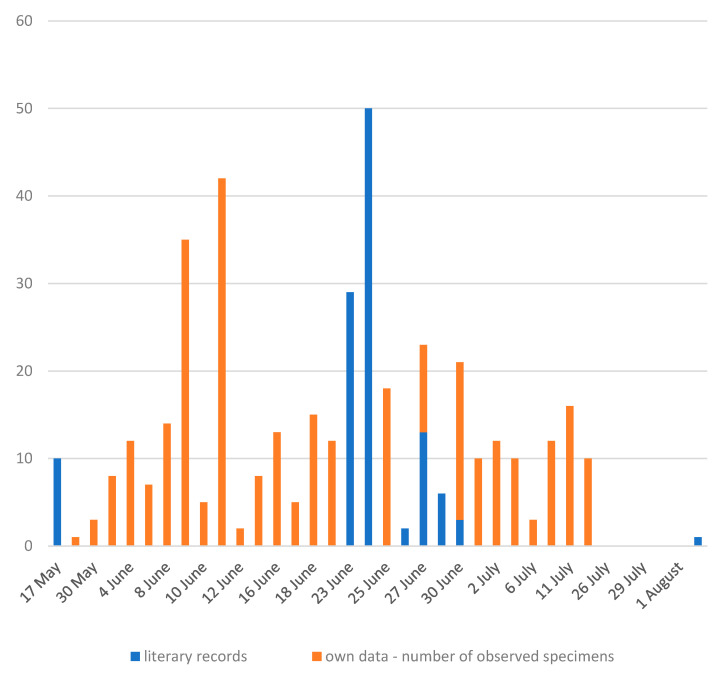
Flight time of the imagines *Cordulegaster buchholzi* in the Cyclades Islands (Greece) (data from years 2010–2023, literary records [1,4,6,7,9,15]).

**Table 1 insects-14-00896-t001:** Description of watersheds with localities with *Cordulegaster buchholzi* records on the Cyclades Islands in Greece (localities are sorted upstream).

Island	Number of Watershed	Locality Name (Cadastral Territory)	Code of Locality	Altitude (m a.s.l.)	Geographical Coordinates (N/E)		Larvae Density (per 1 m^2^ of Sediment)
Andros	I.	Apikia—Vorimi Spilio—under masiv of Petalo Mt.	Ia.	573	37°50′14.93″	24°53′4.53″	1.8
			Ib.	564	37°50′06.88″	24°52′59.14″	10.0
	II.	Exo Vouni—Dipotama stream	IIa.	405	37°47′26.53″	24°55′48.97″	4.0
	III.	Kaparia—Kornov potamos stream	IIIa.	254	37°46′24.96″	24°54′27.50″	1.3
	IV.	Kato Katakilos (Ateni)	IVa.	115	37°52′41.36″	24°48′54.74″	0.1
	V.	Kato Varidi	Va.	120	37°56′42.92″	24°45′51.14″	12.0
	VI.	Lamyra—Platamos stream	VIa.	94	37°49′35.59″	24°54′42.30″	16.0
			VIb.	100	37°49′32.84″	24°54′35.67″	8.0
	VII.	Arni—Remata—Aspropotamos stream	VIIa.	114	37°52′46.29″	24°50′08.94″	0.2
			VIIb.	130	37°52′38.25″	24°50′09.22″	8.0
			VIIc.	173	37°52′00.63″	24°50′12.30″	3.8
			VIId.	190	37°51′51.77″	24°50′12.81″	2.9
			VIIe.	210	37°51′44.59″	24°50′14.09″	7.3
			VIIf.	216	37°51′45.05″	24°50′17.64″	4.0
			VIIg.	232	37°51′36.17″	24°50′11.23″	40.0
			VIIh.	245	37°51′31.61″	24°50′09.70″	7.1
			VIIi.	268	37°51′25.64″	24°50′07.15″	0.2
			VIIj.	388	37°51′08.84″	24°49′57.48″	26.7
		Arni centre	VIIa.	406	37°51′19.20″	24°50′26.10″	4.0
	VIII.	Stenies	VIIIa.	35	37°51′05.24″	24°55′31.29″	10.0
	IX.	Vourkoti	IXa.	670	37°51′25.09″	24°53′09.27″	3.3
	X.	Zaganiaris—Megalos Potamos stream	Xa.	256	37°47′36.48″	24°53′10.50″	3.7
Tínos	XI.	Aetofolia	XIa.	137	37°37′08.97″	25°06′46.54″	21.3
	XII.	Koris Pirghos	XIIa.	402	37°37′25.48″	25°05′16.97″	6.7
	XIII.	Tarampados	XIIIa.	314	37°34′51.34″	25°08′36.64″	0.2
Náxos	XIV.	Abram—Milon Potama	XIVa.	220	37°09′43.26″	25°30′08.93″	8.9
	XV.	Aghia	XVa.	199	37°11′08.91″	25°31′15.24″	20.0
	XVI.	Apollonas—Potamos stream	XVIa.	113	37°09′51.01″	25°32′42.64″	0.4
			XVIb.	205	37°09′00.38″	25°32′37.74″	0.1
	XVII.	Koronida	XVIIa.	455	37°08′31.76″	25°31′54.84″	0.7
			XVIIb.	680	37°07′59.08″	25°31′40.31″	62.2
	XVIII.	Skeponi	XVIIIa.	251	37°08′32.71″	25°29′40.63″	0.4

## Data Availability

Data are contained within the article.

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
