# Peer review of "Occurrence and Population Density of the Endemic Species Cordulegaster buchholzi (Anisoptera: Cordulegastridae) on the Cyclades Islands in Greece"

_insects, 2023, doi:10.3390/insects14110896_

Round 1
Reviewer 1 Report
Comments and Suggestions for Authors
A very valuable paper about a poorly researched species. I do not have any serious general comments, and numerous detailed comments can be found in the PDF file.
I suggest that the authors, when writing about the area of the species in question, specify what type of area they mean, i.e. AOO or EOO? Knowing approximately the size of this area, they can then immediately suggest a specific category of threat to the species. There are strict IUCN guidelines for this.
I also assume that the authors have data suitable for performing multivariate analysis (CCA / RDA / PCA). Then we would find out which of the complex of factors creating the species' habitat niche are the most important. However, perhaps these data are not yet sufficient? This is where the authors have to express their opinion.

Author Response
I thank the opponent for all the detailed comments.
Most of the comments have been incorporated into the text. Some new literary sources have also been added.
A chapter on Conservation status of IUCN was written separately, where the criteria for inclusion in IUCN categories were taken into account.
Image captions have been modified.
Multivariate analysis (CCA / RDA / PCA) could be processed even now, however, for the classification of occurrence factors, it would be advisable to also have data from the part of the streams without the occurrence of the species (which we currently have only to a limited extent and we are working on expansion).
It would be possible to include one larva in a figure, however, to show the characters, it would be necessary to take detailed macro photographs (which will be the subject of a separate work).
Reviewer 2 Report
Comments and Suggestions for Authors
A well conducted research project, well recorded in the paper. I attach a copy of the review manuscript with highlighted suggested corrections (mainly to English expression). As a result of my corrections in three places lines have become superimposed. If those are incorporated the paper is worthy of publication. The specific names in lines 190-193 and 208-211 should be italicized.

In general the English is good, but there are many places where it can be improved.
Author Response
I thank the opponent for all the detailed comments.
Most of the comments have been incorporated into the text. Some new literary sources have also been added.
A chapter on Conservation status of IUCN was written separately, where the criteria for inclusion in IUCN categories were taken into account.
English was modified according to comments and checked.
Latin names that are written in italics are unfortunately always changed when converting to pdf format. In the original .doc file they are written in italics.